# Automated Classification of Auroral Images with Deep Neural Networks

**Zhiyuan Shang** [1,2], **Zhonghua Yao** [1,2,*], **Jian Liu** [3,4], **Linli Xu** [5,6], **Yan Xu** [1,2], **Binzheng Zhang** [7,8,9], **Ruilong Guo** [10,11] **and Yong Wei** [1,2]

1 Key Laboratory of Earth and Planetary Physics, Institute of Geology and Geophysics, Chinese Academy of Sciences, Beijing 100000, China
2 College of Earth and Planetary Sciences, University of Chinese Academy of Sciences, Beijing 100000, China
3 School of Nuclear Science and Technology, University of Science and Technology of China, Hefei 230026, China
4 Advanced Algorithm Joint Lab, Shandong Computer Science Center, Qilu University of Technology, Jinan 250014, China
5 Anhui Province Key Laboratory of Big Data Analysis and Application, School of Computer Science and Technology, University of Science and Technology of China, Hefei 230026, China
6 Anhui Province Key Laboratory of Big Data Analysis and Application, School of Computer Science and Technology, University of Science and Technology of China, State Key Laboratory of Cognitive Intelligence, Hefei 230026, China
7 Department of Earth Sciences, The University of Hong Kong, Hong Kong SAR, China
8 Laboratory for Space Research, The University of Hong Kong, Hong Kong SAR, China
9 High Altitude Observatory, National Center for Atmospheric Research, Boulder, CO 80301, USA
10 Laboratory of Optical Astronomy and Solar-Terrestrial Environment, Institute of Space Sciences, School of Space Science and Physics, Shandong University, Weihai 264200, China
11 Laboratory for Planetary and Atmospheric Physics, STAR Institute, Université de Liège, Liège 4000-4032, Belgium
* Correspondence: z.yao@mail.iggcas.ac.cn

**Abstract:** Terrestrial auroras are highly structured that visualize the perturbations of energetic particles and electromagnetic fields in Earth's space environments. However, the identification of auroral morphologies is often subjective, which results in confusion in the community. Automated tools are highly valuable in the classification of auroral structures. Both CNNs (convolutional neural networks) and transformer models based on the self-attention mechanism in deep learning are capable of extracting features from images. In this study, we applied multiple algorithms in the classification of auroral structures and performed a comparison on their performances. Trans-former and ConvNeXt models were firstly used in the analysis of auroras in this study. The results show that the ConvNeXt model can have the highest accuracy of 98.5% among all of the applied algorithms. This study provides a direct comparison of deep learning tools on the application of classifying auroral structures and shows promising capability, clearly demonstrating that auto-mated tools can help to minimize the bias in future auroral studies.

**Keywords:** aurora; machine learning; CNNs; transformer; transfer learning

## 1. Introduction

Solar wind particles are high-speed flows with magnetic fields that can blow the Earth and compress the Earth's magnetic field in the dayside and extend it in the nightside, forming the terrestrial magnetosphere. Energy and charged particles are gradually stored in the magnetosphere, which are released from time to time, powering auroras in the ionosphere and upper atmosphere. Auroras are the consequence of the coupling of the magnetosphere and ionosphere. Based on the auroral observations from the American continent and Antarctica, Akasofu established the first connection between the auroral morphology and the dynamic processes [1] and developed an auroral substorm model to

describe the evolution of the aurora, which has two main phases: the expansion phase and the recovery phase. The aurora changes from a calm to an active state and back to a calm state. This evolutionary pattern was soon confirmed by satellite observations [2,3]. The model was subsequently refined into the three phases of "growth", "expansion", and "recovery" that are now widely used [4]. The rapid release of energy stored in the magnetosphere corresponds to the substorm expansion phase, which is featured with a rapid expansion of the auroral activity zone to form a bright bulge in space in which most of the auroral arcs show a distinct curtain-like fold and gradually break up and die out in rapid motion. At the same time, new auroral arcs continue to emerge, causing the bulge zone to expand in both longitude and latitude. The westward and polar expansion of the aurora is particularly pronounced. The recovery phase usually last for tens of minutes to hours, where the magnetosphere then returns to pre-substorm state.

Since 1892, when German physicist and astronomer Otto Rudolf Martin Brendel took the first photographs of the Northern Lights, today millions of images of the aurora borealis have been taken every year in the North and South Polar Regions by scientific cameras in space and on the ground; moreover, citizen photographers can also provide highly valuable photos that can help us to make scientific discoveries (https://www.science.org/doi/10.1126/sciadv.aaq0030, accessed on 14 March 2018). The mega-dataset of auroral images can greatly improve our understanding of auroral processes from global features to micro-details. Meanwhile, the analysis of such a large dataset strongly requires automated tools. In recent years, machine learning has rapidly developed and has been applied in many fields, including in face detection, speech recognition, image classification, and medical diagnosis. While the application of machine learning in the field of auroras is relatively rare, the automatic classification of a large number of auroral images can not only reduce our workload but also largely eliminate the biases caused by human factors.

The machine classification of auroras is a challenging task because auroras rapidly change and perform multiple structures in a short period of time; furthermore, there is currently no clear consensus on the classes of auroras. Even for manual classification, it is often ambiguous. Nevertheless, some automated classification techniques have been developed; for example, Syrjäsuo and Pulkkinen [5] made the first attempt to classify images by determining the shape skeleton in each auroral image, which was subsequently used in the identification of auroral arcs [6,7]. Then, using the K-nearest neighbors (KNN) model [8] for aurora tracking, the authors examined the existence of an aurora with an accuracy of ~90%, and they determined the occurrence rate for the auroral arcs, patches, and omega bands [9]. Yang et al. used a hidden Markov model to classify aurora data; the authors obtained the occurrence distributions of four kinds of aurora [10], and a further work was to use the labels of no aurora, aurora, and cloudy and use a support vector machine (SVM) to classify with 90% accuracy [11]. However, due to the fact that the above-mentioned algorithm can only achieve a few classes of classification and due to the poor accuracy, the application is still limited. There has also been the use of a more classic convolutional neural network (CNN) to classify images. Yang et al. used the AlexNet architecture to extract multi-scale contextual features [12]. The improved CycleGAN algorithm was used to extract the key local structures of all sky auroral images, and the accuracy can reach 92% [13].

Clausen and Nickisch [14] labeled 5824 auroral images from multiple all-sky imagers of the Time History of Events and Macroscale Interactions during Substorms (THEMIS) spacecraft with clear/no aurora, cloudy, moon, arc, diffuse, and discrete. Through the pre-trained neural network, the 1001-dimensional feature vector is automatically extracted from the auroral images, and the label and feature vector are used together to train the ridge classifier. The accuracy for identifying the six classes of aurora is 82%, and it resulted in a 96% rate for discriminating between aurora and no-aurora/clear. Later, transfer learning was introduced into aurora image classification [15]. Kvammen [16] attempted six deep neural network architectures and compared them with traditional machine learning classification algorithms. The results show that deep neural networks are generally better

than KNN and SVM methods, among which the ResNet-50 architecture has been shown to have the best performance and accuracy at 92%. Nanjo [17] used 5,530,796 auroral images taken in Tromsø, Norway to further train the ResNet-50 network and achieved 93% accuracy, obtaining the annual, monthly, and UT variations of the aurora occurrence frequency. Guo et al. [18] compared different CNN architectures and different layers using mesoscale images of auroral structures to test the best model, and the authors observed the time-series of auroral evolution through automatic recognition.

Following the previous research work, the main purpose of this study is to attempt new algorithms to further improve the accuracy of classification and generalization ability of the model. As of today, convolutional neural networks (CNNs) have performed well in the field of computer vision and can satisfy image classification tasks in many fields. In particular, the ConvNeXt algorithm proposed by Liu et al. [19] this year has further improved the upper limit of the convolutional neural network. By testing the ConvNeXt model, we have found that the average accuracy of aurora classification can reach up to 98.5%, which is satisfactory in many topics in the auroral field. In addition, the transformer model is relatively popular in the field of natural language processing, and it has also been extended to the field of computer vision. We have tested it with MobileViT and the swim transformer model, and the average classification accuracy was 93.7% and 94.5%, respectively. Before training the model, we manually marked 4751 images taken by THEMIS all-sky imagers at different times and labeled them with different labels, such as arc, block, border, cloudy, diffuse, discrete, faint, moon and others.

## 2. Methodology

Our goal was to train a model to automatically classify aurora based on image features. It should be noted that our model is easily transferable to other aurora datasets. The algorithms tested below are all supervised learning. Once the training set is determined, the model can be trained, and then new auroral images can be analyzed.

### 2.1. Aurora Labels

The classes of auroral structures are largely based on previous literature but are not identical to a single study; six of the types follow the classification criteria of Clausen and Nickisch [14]. In order to better study the continuous time aurora sequence changes and to eliminate unidentifiable and polluted auroral images, we introduce three new classifications: border, block, and others. The specific classification standards are in Figure 1.

| Index | Label | Explanation |
|:---:|:---:|:---|
| 0 | arc | show one or multiple bands, curve |
| 1 | block | the auroral image is highly contaminated so that we could not get useful information about the auroral morphology. |
| 2 | border | Images with auroral emission only at the edges of the image are labelled as edge aurorae, which can be any of the above classes of aurorae, but information about such aurorae is limited by the insufficient number of bright pixels and the uncertainty of the form of the aurorae outside the image frame. |
| 3 | cloudy | the images is dominated by clouds |
| 4 | diffuse | typically with fuzzy edges |
| 5 | discrete | the images show auroral forms with well-defined, brighter than the arc auroras because it has a higher energy. |
| 6 | faint | no aurora |
| 7 | moon | the image is dominated by light from the dot that like moon |
| 8 | others | ambiguous image |

**Figure 1.** Categories of auroral classification and criteria for classification.

We need to point out that the labelling of auroral images would inevitably introduce ambiguities. Although we have adopted six of the same categories from previous literature, the label for each aurora image is not completely the same, which is because some auroral images may show complex structures and thus can be classified into different categories by different researchers. For example, the patchy described by Syrjäsuo and Donovan [9] and Kvammen [16] are in general agreement with our diffuse type, whereas in our classification, auroral images showing both the discrete and diffuse features are classified as the discrete auroral type. Although our training set is still somehow subjective, the model can be well-trained as long as we can keep a consensus in the classification.

### 2.2. Pre-Processing of Images

The images we use for the training dataset are all from FSIM, RANK, and ATHA of the widely used and readily available THEMIS all-sky imager network. We selected the auroral images of the above three stations in February of 2011, 2014, 2016, and 2019. All THEMIS ground stations use the same lens to capture visible auroras in the wavelength range of 400–700 nm (e.g., white light), and it is capable of taking aurora snapshots every 3 s.

Because the aurora image is relatively complex, we first normalize its gray value to the [0, 1] range and then use gamma correction ($O_{(r,c)} = I(r,c)^\gamma, 0 \leq r < H, 0 \leq c < w$) to enhance the contrast of the aurora image. Figure 2a–i contains nine auroral images that have been preprocessed in this way.

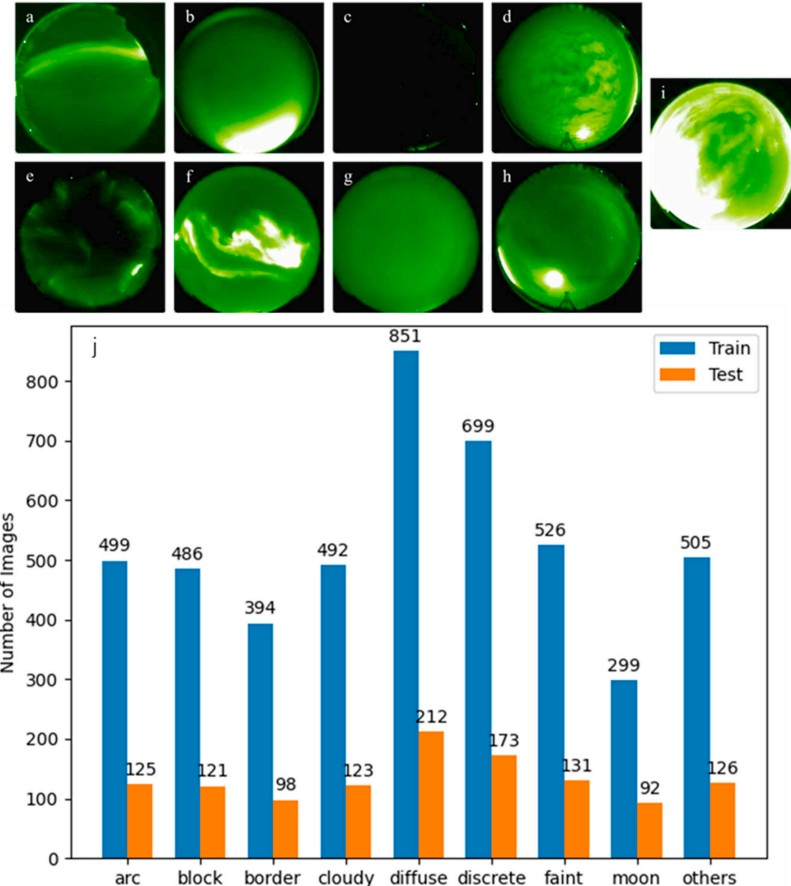

**Figure 2.** In panel (**a**–**i**), we show examples of processed auroral images for the categories arc, block, border, cloudy, diffuse, discrete, faint, moon, and others. Panel (**j**) represents the number of training and test sets for different aurora subclasses.

We selected 5952 pictures containing all of the above nine phenomena from 60,000 auroral images and randomly divided them into a training set and test machine according to the ratio of 8:2, with 4751 pictures being in the training set and 1201 pictures being in

the test set. Because different aurora subclasses have different durations and occurrences, we use the total duration of each subclasses to nominalize the time gap to select images from the dataset to ensure that the typical features of different auroral types were covered. For example, if one type of aurora has a longer duration than another one, the even time gap is also proportionally longer; it may be 30 s, 1 min, or more. As such, there is a lot of flexibility in this way, as such a selection can guarantee representative auroral datasets for better training. The number of images in each subclass in the training and test sets is shown in Figure 2j.

*2.3. Neural Network Architecture*

This study used two convolutional neural networks and two transformers for comparison. The convolutional neural network is a multi-layer supervised learning neural network. The convolutional layer and pooling sampling layer of the hidden layer are the core modules that are used to realize the feature extraction function of the convolutional neural network. The network model reversely adjusts the weight parameters in the network layer by using the gradient descent method to minimize the loss function, which improves the accuracy of the network through frequent iterative training [20]. The low hidden layer of the convolutional neural network is alternately composed of convolutional layers and maximum pooling sampling layers, and the high layer is the hidden layer and logistic regression classifier of the fully connected layer corresponding to the traditional multi-layer perceptron. The input of the first fully connected layer is the feature image obtained by the feature extraction of the convolutional layer and the subsampling layer. The last output layer is a classifier that can use logistic regression, softmax regression, or even a support vector machine to classify the input image. The transformer model was firstly proposed by Google in 2017 [21]; it uses the self-attention structure to replace the recurrent neural network (RNN) network structure that is commonly used in natural language processing (NLP) tasks and can achieve parallel computing. The transformer model is mainly composed of an encoder and a decoder. Its multi-head attention (multi-head attention, MHA) can capture richer feature information, and it is more computationally efficient than CNNs.

A number of neural networks have been used for image classification, and this study evaluated the performance of four widely used neural networks (summarized below and described in detail by Supplementary Materials) for aurora classification work.

- **ResNet**

ResNet [22] is a residual network, which we can understand as a sub-network that is stacked to form a very deep network. The deeper the network is, the more information we can capture and the richer the features. On the other hand, a deeper network also means that more gradient explosion and gradient disappearance can occur. ResNet innovatively proposes a jump connection to solve these problems. The ResNet-34 and ResNet-50 classifications were tested in our study.

- **Swim Transformer**

In recent years, computer vision has entered the "roaring" 2020s, starting from the introduction of the vision transformer (ViT) model [23], which can quickly surpass the CNN model and achieve SOTA recognition performance. The swim transformer model [24] uses sliding windows to endow the model with a linear computational complexity, which improves the information exchange between windows through cross-window connections and ultimately improves the performance of the model in the applications of image classification, object detection, and instance segmentation.

- **MobileVit**

MobileViT is a lightweight visualization translator network for mobile devices [25]. CNNs can only acquire local features, whereas transformer models can acquire global features; however, transformer is a heavy model. MobileVit combines the advantages of

both and building a MobileVit combines the advantages of both, resulting in a lightweight and low-latency network.

- **ConvNeXt**

ConvNeXt is a pure CNN network [19]. Liu et al. improved on the structure of the ResNet-50 network, which was modelled on that of the swim transformer. The results are very good and have surpassed the swim transformer in terms of performance.

In this study, we have used transfer learning in the process of using the above algorithm, which has the advantage of better initial performance of the model and does not require a large amount of training data.

## 3. Results

To qualitatively measure how well our model does in the classification task, we use the TP (true positive), TN (true negative), FP (false positive), and FN (false negative) parameters, where positive refers to the predicted classification subclasses of the aurora image, negative (N) indicates all other subclasses, and true (T) and false (F) indicate whether the prediction is correct or incorrect. There are also precision, recall, and F1 to consider, as well as the confusion matrix and the average accuracy of the model.

Precision is the proportion of data for which the prediction is correct and the true value is correct. Precision is defined as follows:

$$Precision = \frac{true\ positive}{true\ positive + false\ positive}$$

Recall is the number of data that can be correctly predicted out of all of those whose true value is correct, which is defined as follows:

$$Recall = \frac{true\ positive}{true\ positive + false\ negative}$$

The F1-score combines the precision and recall of a classifier into a single metric by taking their harmonic mean. It is primarily used to compare the performance of two classifiers. Suppose that classifier A has a higher recall and classifier B has a higher precision. In this case, the F1-scores for both the classifiers can be used to determine which one produces better results. The maximum value of F1 is 1, and the minimum value is 0. The larger the value, the better the performance. it is defined as follows:

$$F1 = 2 * \frac{precision * recall}{precision + recall}$$

The three parameters above measure the effectiveness of the classification of each aurora subclass in addition to the overall accuracy of the model:

$$Accuracy = \frac{TP + TN}{TP + FN + FP + TN}$$

### 3.1. Comparison of Different Models

In order to visually compare the performance of the four models in aurora classification, the confusion matrix of the different models is shown in Figure 3. ResNet-50 has an accuracy of 93.8%, in which most of the aurora subclasses are correctly classified but there are some problems in the discrimination of diffuse and discrete, which are also challenging for the human eye to recognize. The accuracy of ConvNeXt is 98.5%, which is the best performance among all models. Due to the limitations of the computer conditions, we have used the ConvNeXt-T model in our tests, which would have performed better if we had used the ConvNeXt-XL model. The accuracy of the swim transformer (Swim-T) model is 94.5%; for MobileVit (MobileVit-XXS), it is 93.7%, and it is a lightweight network with better results and better generalization ability when using the same parameters. The above

models (except for ResNet-50) were chosen with smaller parameters, and the classification performance will be further improved if a larger model is chosen.

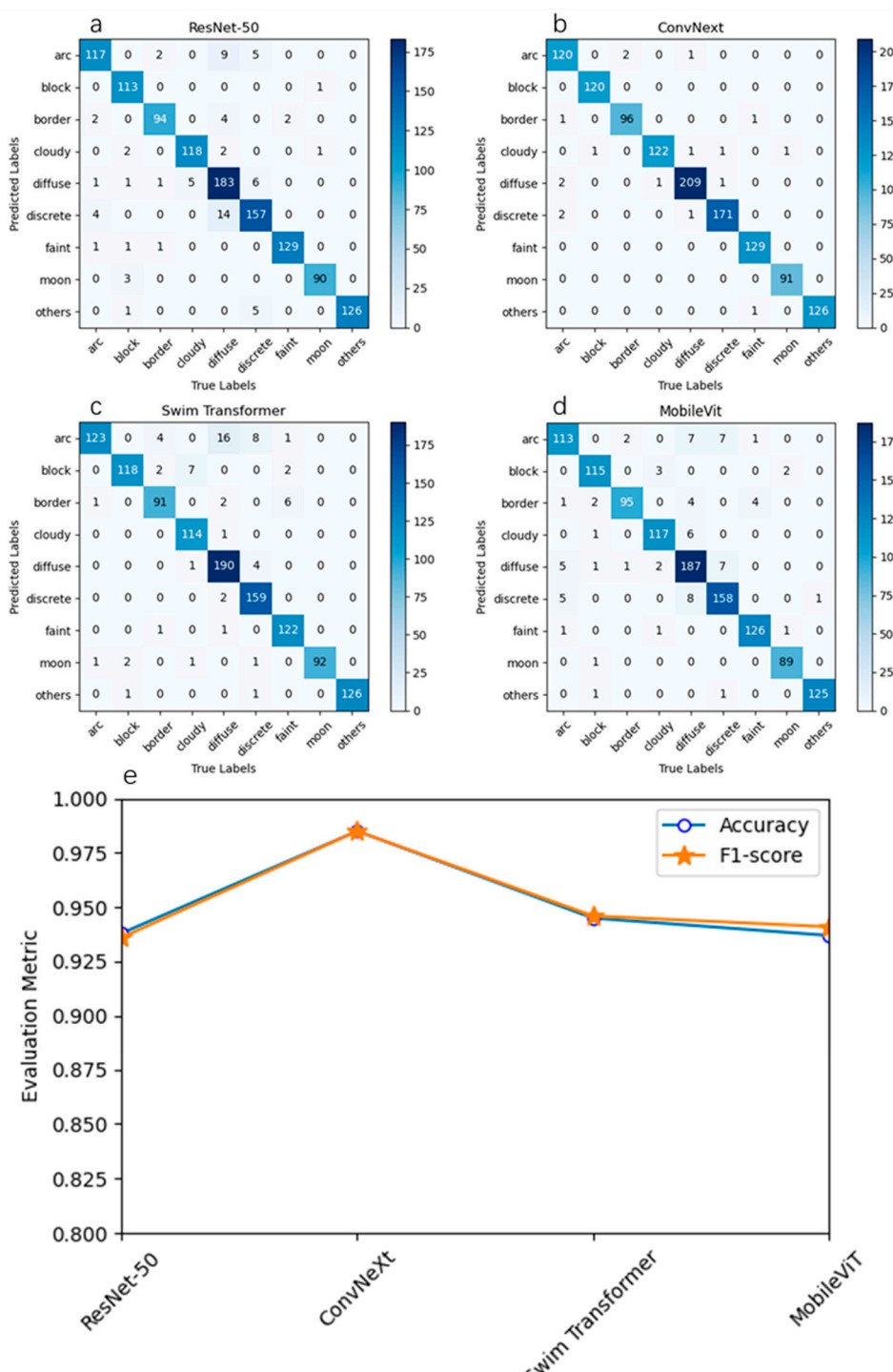

**Figure 3.** Confusion matrix of the four neural networks on the test set. (**a**) ResNet-50; (**b**) ConvNeXt; (**c**) swim transformer; (**d**) MobileVit; (**e**) comparison of accuracy and F1-scores of the four algorithms.

Furthermore, Table 1 provides the precision, recall, and F1 scores of the four models for the different auroral subclasses. The performance of the different models for the different subclasses also has its own advantages. For example, ConvNeXt performs well in distinguishing between discrete and diffuse, and swim transformer is best at identifying moons, so that when we focus on different subclasses, we can choose different algorithms.

**Table 1.** Precision, recall, and F1 scores for each subclass of the test set using (a) ResNet-50, (b) MobileVit, (c) swim transformer, and (d) ConvNeXt neural network models.

| a | ResNet-50 | | | b | MobileVit | | |
|---|---|---|---|---|---|---|---|
| | Precision | Recall | F1 | | Precision | Recall | F1 |
| arc | 0.88 | 0.936 | 0.907 | arc | 0.869 | 0.904 | 0.886 |
| block | 0.991 | 0.934 | 0.962 | block | 0.958 | 0.95 | 0.954 |
| border | 0.922 | 0.959 | 0.94 | border | 0.896 | 0.969 | 0.932 |
| cloudy | 0.959 | 0.959 | 0.959 | cloudy | 0.944 | 0.951 | 0.947 |
| diffuse | 0.929 | 0.863 | 0.895 | diffuse | 0.921 | 0.882 | 0.901 |
| discrete | 0.897 | 0.908 | 0.899 | discrete | 0.919 | 0.913 | 0.916 |
| faint | 0.977 | 0.985 | 0.902 | faint | 0.977 | 0.962 | 0.969 |
| moon | 0.968 | 0.978 | 0.981 | moon | 0.989 | 0.967 | 0.978 |
| others | 0.955 | 1 | 0.977 | others | 0.984 | 0.992 | 0.988 |
| c | Swim Transformer | | | d | ConvNeXt | | |
| | Precision | Recall | F1 | | Precision | Recall | F1 |
| arc | 0.809 | 0.984 | 0.888 | arc | 0.976 | 0.96 | 0.97 |
| block | 0.915 | 0.975 | 0.944 | block | 1 | 0.992 | 0.996 |
| border | 0.91 | 0.929 | 0.919 | border | 0.98 | 0.98 | 0.98 |
| cloudy | 0.991 | 0.927 | 0.96 | cloudy | 0.968 | 0.992 | 0.98 |
| diffuse | 0.974 | 0.896 | 0.933 | diffuse | 0.981 | 0.986 | 0.983 |
| discrete | 0.988 | 0.919 | 0.952 | discrete | 0.983 | 0.998 | 0.985 |
| faint | 0.984 | 0.931 | 0.957 | faint | 1 | 0.985 | 0.992 |
| moon | 0.948 | 1 | 0.973 | moon | 1 | 0.989 | 0.994 |
| others | 0.984 | 1 | 0.992 | others | 0.992 | 1 | 0.996 |

*3.2. Test Case (ConvNeXt)*

To qualitatively test whether our network can perform well on other auroral images, we experimented with the best-performing ConvNeXt model. We used 180 consecutive images taken at the INUV station from 08:00:00 to 09:00:00 on 15 November 2015; Figure 4 shows the classification results. It can be seen that initially from the arc, it gradually evolved into discrete, then into diffuse, and then the two alternatingly appeared. There is a good correspondence with the captured auroral images and mosaic diagrams, indicating that our model has a good classification performance, which lays the foundation for subsequent research on the evolution of the aurora.

In the middle panel and the aurora plot at the corresponding moment above, because the ConvNeXt model identifies the auroral subclasses with high accuracy (0.98 or 0.99 for all but arc's F1 score of 0.97), the probabilities of the incorrect classes in the predicted probability stacked plots above are all mostly close to zero.

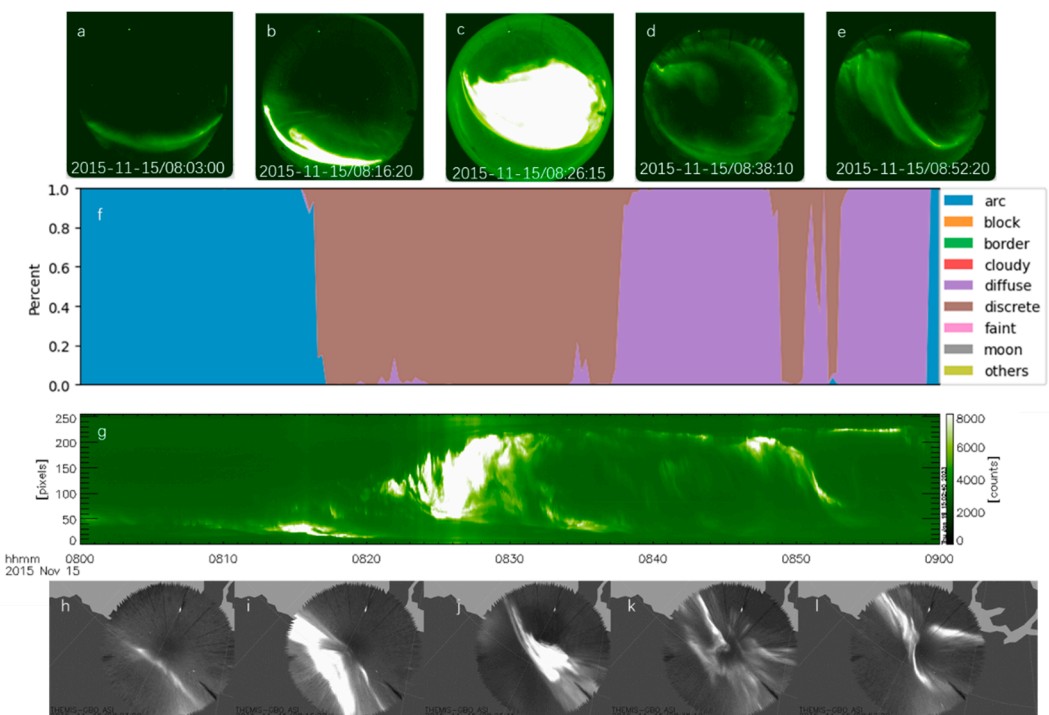

**Figure 4.** Auroral images from 08:00:00–09:00:00 on 15 November 2015. (**a–e**) Typical FOV images for the intervals to be discussed throughout the event; in order, they are arc, discrete, discrete, diffuse, and diffuse. (**h–l**) Mosaic images for the corresponding moments on the INUV station. (**f**) is the classification probability of the ConvNeXt network for these consecutive auroral images; (**g**) is the keogram plot.

## 4. Discussion and Summary

In this study, we tested multiple deep-learning tools in the classification of auroral images from ground cameras. ConvNeXt outperformed the transformer model; in general, the ResNet-50, swim transformer, MobileViT, and ConvNeXt models all had high accuracy (>93%). Thus, aurora classification is a suitable job for DNNs. Regarding the highly challenging task of distinguishing discrete from diffuse and curved from discrete, the ConvNeXt model is highly capable. As a comparison, the clustering method [26] in unsupervised learning only has two classes, which are perhaps insufficient in the classification of auroras that is driven by a mixture of many physical processes.

We selected 5952 auroral images from the THEMIS All-Sky Imager and manually labeled nine categories of aurora: arc, block, boundary, cloudy, diffuse, discrete, faint, moon, etc., and then used convolutional neural networks and the transformer model to train it. Transformer is our first attempt at aurora classification, and it has a higher potential than CNNs in future machine learning investigation of auroral physics. Several of our models perform well after training, especially the ConvNeXt model, which can achieve a 98.5% accuracy. It can also be applied to other aurora datasets, and it will help auroral scientists study the patterns of auroral changes from a new perspective.

This study uses the PyTorch architecture, and it is easy to make changes to the labels used here. Of course, there can be more aurora subclasses if needed for some specific research purposes. In a future work, we will consider additional aurora subclasses to capture fine auroral structures that can indicate fundamental plasma processes, for example, for aurora in sunlight [27,28], Strong Thermal Emission Velocity Enhancement (STEVE) [29,30], omega bands [31], etc. We will also apply our model to the study of the incidence of aurora under different solar wind conditions and the evolution of aurora under different geomagnetic conditions, which are important information in space weather considerations, which cannot be easily performed by eye-based classification.

**Supplementary Materials:** The following supporting information can be downloaded at: https://www.mdpi.com/article/10.3390/universe9020096/s1, Figure S1: CNN structure of ResNet-50; Figure S2: The architecture of a Swin Transformer; Figure S3: ConvNeXt algorithm improvements and results.

**Author Contributions:** Conceptualization, Z.Y. and J.L.; methodology, Z.Y. and J.L.; software, L.X.; writing—original draft preparation, Z.S. and Z.Y.; writing—review and editing, Y.X., R.G., B.Z. and Y.W. All authors have read and agreed to the published version of the manuscript.

**Funding:** The National Science Foundation of China (grant no. 42074211) and the Key Research Program of the Institute of Geology & Geophysics CAS (grant no. IGGCAS-201904).

**Data Availability Statement:** The THEMIS all-sky data are available through the THEMIS website at http://themis.ssl.berkeley.edu. PyTorch is available at http://pytorch.org accessed on 31 October 2016). ConvNeXt is available at https://github.com/facebookresearch/ConvNeXt (accessed on 2 March 2022). Swim Transformer is available at https://github.com/microsoft/Swin-Transformer (accessed on 17 August 2021). MobileViT is available at https://github.com/apple/mL-cvnets (accessed on 5 October 2021).

**Acknowledgments:** We thank the entire THEMIS team for the data. Part of the data processing uses SPEDAS [32]; the data are available through the THEMIS website at http://themis.ssl.berkeley.edu (accessed on 30 November 2022).

**Conflicts of Interest:** The authors declare no conflict of interest.

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
