# Peer review of "Automated Classification of Auroral Images with Deep Neural Networks"

_universe, doi:10.3390/universe9020096_

Round 1
Reviewer 1 Report
The paper "Automated Classification of Aurora Images with Deep Neural Networks" by Shang et al., studied several deep-learning techniques for classifying auroral images. The methodology are well presented; the results are robust as shown by the confusion matrix.
Author Response
Thank you for supporting us in publishing research reports and giving positive feedback on our main results.
Reviewer 2 Report
The article is interesting. Minor flaws due to the lack of details:
1) There is no interesting for readers comparison of the network training time when working on typical equipment.
2) It is not clear how many times each model was run. If only once, then it is incorrect to compare the success of models, because the spread between implementations can change the result by several percentage points where there is a difference. In this case, the conclusions should be more accurate.
I think it will be easy for the authors to add this information to the existing text.
Author Response
Thank you very much for your constructive comments/suggestions on our previous manuscript. We have made corresponding revisions following the reviewer’s comments. We believe that the revised manuscript is now suitable for a publication. Thanks for reconsider our revised manuscript.
1. Thank you for your suggestion. We have removed the comparison part of network training time from the original text and put it in the supplementary file, in case some readers are interested in the information.
2. Each of our models is trained with three different random seeds to obtain its mean, and during each run, the training set and test machine were randomly disrupted and redistributed according at 8:2 to ensure accurate results. The code is released and described in the revised manuscript.
Reviewer 3 Report
The study tests new more advanced CNN methods for auroral image classification. A new network called ConvNeXt appears to perform extremely well on this task. This is an important topic because the amount of image data collected these days is so much that most of the information gets lost without automatic analysis tools. The manuscript is well-written and easy to read. I have, however, some major concerns about the study that must be addressed before the manuscript can be considered for publication.
1. The image dataset must be thoroughly described. That includes at least station, wavelength, resolution, and cadence. Furthermore, the image sets used for training and testing also need to be described, including at least their distribution in dates, times and magnetic activity, and in particular their cadence and overlap. Several earlier studies have shown good results in AI until it has turned out that the images in training and testing sets are temporarily too close to each other. That this is not the case here must be demonstrated.
2. The methods need to be more thoroughly described. The current version of the architecture section is very general. CNN implementation comes with a number of free parameters, which need to be described in publications for the repeatability of the studies. It is also recommended to publish the codes in GitHub, for instance, and also publish the labelled data. Less experienced readers would also want to see a human understandable summary of what is different between the different methods, what are their advantages and disadvantages, and what do the difference scores actually mean.
3. Labelled data shown in this study is very limited and some of the labels are not well-described. Examples of unclear definitions: Block - unclear if this class only relates to the emission coverage or perhaps also the brightness; Border - unclear if this class only relates to the brightness of the emission or also its location with respect to the horizons; Diffuse - unclear how motion can be assessed based on individual images, also unclear how well this class is labelled as the example images show sharp edges; Discrete - unclear why structures in this class should be limited to bands, is that not what Arcs are.. Furthermore, the qualitative test example event should surely include all the different classes rather than just 3 out of 9 classes. There is no shortage of data. In figures displaying sample images it should be clear what the assigned class is.
4. The amount of data in training and testing is not enough. There have been earlier studies with only some thousands of images, but all most recent studies use an order of magnitude more data. That should be done here too, because there is no shortage of data. Too small datasets become convoluted and overfitted, which would explain the extremely high success rate of the best performing method in this study. Without larger datasets in training and testing it is not credible that the change of method (albeit a newer one) increases the success rate from about 93% up to 98%.
Round 2
Reviewer 3 Report
Some things have been improved in this manuscript but some of the most fundamental issues persist. These are summarized below and must be fully incorporated before the full value of the study can become clear and the presentation can become transparent.
Data description:
The authors’ response promises improved image data description but little of that is seen in the revised text. Three THEMIS camera stations are mentioned on line 140, but not the years, filters, cadence or the image separation between the training and testing sets. It is very important to be thoroughly clear about how the training and testing sets have been chosen from the huge amount of data, and how well the two sets are separated, and how is it ensured that they cover a large variety of auroral and sky conditions. What is an even time gap the authors refer to? What are the typical auroral morphologies the authors refer to?
Image classes:
The descriptions have improved slightly but much of the clarifications in the response text have not made it to the revised manuscript. Some of the remaining unclear issues:
Block — is the contamination only related to non-auroral sources?
Border — The figure 1 description is not informative enough.
Diffuse — If this is a class of patchy aurora, then it needs to be called patchy. If the typical diffuse aurora has fuzzy edges, the example images must be chosen to illustrate this. The example images do seem to have aurora with rather sharp boundaries, which will probably change drastically from one image to another, which does not comply with the definition of “little motion”. I recommend the authors to familiarize with the recent review article on diffuse and pulsating aurora by Nishimura et al. in Space Science Reviews, 2020.
Discrete — The Figure 1 description is cut off. The authors’ response refers to energy, but it is unclear how the energy is related to the image features, and this is also not discussed in the article. It also remains unclear if the classes of Arc and Discrete are meant/known to have a lot of overlap.
Others — Is this class also solely non-auroral contamination? If so, how does it differ from the Block class?
The description of the different classes is inevitably subjective, as the authors point out. This is exactly why the descriptions need to be done as detailed and carefully as possible, and with sufficient number of example images. It is understandable that one single time period of aurora may not display all the different classes, but it is therefore important to include more examples so that all the classes are being displayed with sample images, and that the accuracy of the detection of the different classes is being demonstrated.
Figure 5: Station names are missing from the displayed images and the fields-of-view of the relevant images need to be fully visible rather than cut off by the edge of the map.
Methods:
While the supplementary material is very helpful, the method comparison in Figure S3, or similar information, needs to be implemented in the article text to justify the benefit of ConvNeXt over the other methods. In addition to publishing the methods themselves, it would be highly recommended to also publish the manually labelled training set of images.
Round 3
Reviewer 3 Report
Much of the descriptions for data and classes have been improved a lot and are largely very clear now. I have a few relatively small but very important requests for the paper before I am happy to recommend publishing.
* Classes: Good descriptions now. The class Others would also deserve an example image to demonstrate how they look like. That class is anyway mentioned in the Figure 2 caption.
* Data description is much better now. The duration selection criterion is a very good idea. However, it is not clear how the time separation has been determined. Are the total durations determined from the training/testing/validation sets? How do you get the total durations before the classification is done? What are they for different auroral types? And consequently, what are the separation times for different types?
* A minor point is that there are some “aurora images” which should be “auroral images” for consistency.
